# The Influence of Steel Fiber Tensile Strengths and Aspect Ratios on the Fracture Properties of High-Strength Concrete

**DOI:** 10.3390/ma12132105

**Published:** 2019-06-30

**Authors:** Won-Chang Choi, Kwon-Young Jung, Seok-Joon Jang, Hyun-Do Yun

**Affiliations:** 1Department of Architectural Engineering, Gachon University, Gyeonggi-do 461-701, Korea; 2Korea Land & Housing Corporation, Gyeongsangnam-do 52852, Korea; 3Korea Infrastructure Safety Corporation, Gyeongsangnam-do 52856, Korea; 4Department of Architectural Engineering, Chungnam National University, Daejeon 305-764, Korea

**Keywords:** steel fiber, fiber content, aspect ratio, toughness index, high-strength concrete

## Abstract

Steel fiber embedded in concrete serves to reduce crack development and prevent crack growth at the macroscopic level of the concrete matrix. Steel fiber-reinforced concrete (SFRC) with high compressive concrete strength is affected primarily by the dimensions, shape, content, aspect ratio, and tensile strength of the embedded steel fiber. In this study, double-ended hook steel fiber was used in SFRC with a concrete compressive strength of 80 MPa. This fiber was used for the study variables with two aspect ratios (64, 80) and tensile strength values up to 1600 MPa. The flexural performance of the SFRC specimens was evaluated using crack mouth open displacement tests, and the test results were compared with code provisions. A modified reinforcement index was also used to quantify the flexural performance based on comparisons with fracture energy.

## 1. Introduction

The addition of steel fiber into concrete mixtures mitigates brittle failures in the concrete matrix. Specifically, the bridging effect of the steel fiber in the concrete mixture, which occurs after cracking, enhances the mixture’s structural behavior in terms of shear strength, flexural strength, ductility, impact resistance, and fatigue [1,2,3]. The distribution of stress throughout steel fiber-reinforced concrete (SFRC) after cracks appear causes the cracks to widen significantly at the macroscopic level of the concrete matrix so that local crack propagation can be controlled. However, an excessive amount of added steel fiber leads to a reduction of the workability and segregation of the mix [4]. Experimental studies found in the literature [5] are limited to steel fiber aspect ratios of 70 and tensile strength values of 1000 MPa. Therefore, for test purposes, fiber content alone as a test variable is not sufficient. The fiber aspect ratio and tensile strength must also be considered. A wider range of design variables, including various aspect ratios and high tensile strengths, is needed. 

Gao et al. (1997) [6] investigated the flexural behavior of high-strength concrete with a range of fiber contents (0%–2.0%) and aspect ratios (46–70). As the fiber content and aspect ratio were increased in the Gao et al. study, the flexural strength improved by 9.6% to 90%, and the experimental results indicated that a fiber content of 1.0% to 1.5% is needed to enhance the flexural behavior of high-strength concrete. Yazıcı et al. (2007) [7] derived similar results to Gao et al. (1997) [6], who showed that the flexural strength increased from 3% to 81% with an increase in the steel fiber content and aspect ratio in SFRC mixtures. The mechanical properties of SFRC with the compressive concrete strength of 50 MPa and variables of different steel fiber contents (0%–1.5%) and aspect ratios (45, 65, 80).

Köksall et al. (2008) [8] also evaluated the flexural performance of concrete with compressive strength values of 40 MPa to 70 MPa and two different fiber aspect ratios (65, 80), as well as fiber contents (0.5%, 1.0%). As in the earlier studies by Gao et al. (1997) [6] and Yazıcı et al. (2007) [7], the flexural strength increased with an increase in fiber content and aspect ratio. However, the flexural strength decreased with an increase in the compressive strength of the concrete. The result was a higher bond strength of the concrete matrix than the tensile strength of the fiber, which caused the fibers in the concrete mixture to rupture. Therefore, Köksall et al. (2012) [9] continued to study the mechanical properties of SFRC with respect to concrete compressive strength and fiber tensile strength. They found that variations of the mechanical properties of SFRC were insignificant when using normal concrete. However, the mechanical properties were highly affected in SFRC with high-strength concrete.

In short, the high tensile strength of the steel fiber used in high-strength SFRC contributes to the mitigation of cracking and to energy dissipation. Related research conducted by the author indicates the necessity for high-performance steel fiber to improve the toughness and flexural strength of high-strength concrete [10]. Studies associated with the mechanical characteristics of high-strength SFRC, which focus on fiber content, aspect ratio, and tensile strength, are needed to ensure good performance without loss of workability due to the excessive fiber content in high-strength SFRC. In this study, we investigated the flexural performance of SFRC by measuring crack mouth opening displacement (CMOD).

## 2. Experimental Program

### 2.1. Material Preparation and Fabrication

In this study, the design compressive concrete strength (*f_ck_*) was 80 MPa, with a water-to-binder ratio of 25% and steel fiber contents of 0.5% and 1.0% (percentage by volume) as variables. Table 1 shows the mix design of the SFRC used in this study. Typical Portland cement Type I (Density: 3.15 g/cm^3^), Crushed aggregate, Fly-ash Density: 2.12g/cm^3^, Blaine: 2976 cm^2^/g)

Figure 1 presents the different dimensions of the double-ended hook steel fiber used in this study. The literature has shown that double-ended hook steel fiber provides excellent performance [11]. In this study, the main variables were the steel fiber aspect ratios and tensile strength values. For the specimens with aspect ratios of 64, two tensile strengths of 1200 MPa and 1600 MPa were used, and for the specimens with the aspect ratio of 80, two tensile strengths of 1000 MPa and 1600 MPa were used. 

In order to evaluate the effects of the properties of the steel fiber on the flexural behavior of the SFRC, nine beam specimens, 150 mm × 150 mm × 550 mm, were prepared according to the European Standard EN-14651 standard [12]. These flexural test specimens were cured in a steel mold for 24 h and then, after removal from the mold, were cured in water at 20 °C ± 1 °C until testing. Prior to testing, the specimens were also sawed to create a 5 mm wide and 25 mm deep notch on the bottom of each specimen, as shown in Figure 2.

The specimen designations used in this study reflect the aspect ratio of the steel fiber, the tensile strength of the steel fiber, and the steel fiber content (its volume fraction). For example, 64-NTS-0.5 indicates that the specimen has a fiber aspect ratio of 64, a normal tensile strength (NTS) (1200 MPa), and 0.5% fiber. 

### 2.2. Test Set-Up

Flexural tests were conducted using a 200 kN universal testing machine (Heungjin, Kimpo, Republic of Korea). Figure 3 shows the test set-up for a CMOD specimen. Center-point loading was applied with a displacement rate of 0.3 mm/min. LVDTs (linear variable differential transformers) were installed to measure the CMOD. 

## 3. Results and Discussion

### 3.1. Flexural Stress CMOD Curves for SFRC Specimens

Figure 4 presents the flexural stress-CMOD curves of the SFRC mixtures with respect to their different steel fiber contents and the tensile strengths of the steel fiber.

The flexural performance of the SFRC was evaluated in accordance with the EN-14651 standard [12]. The bending stress in the linear region and residual bending stress were calculated using Equations (1) and (2), respectively:(1)$$f_{LOP}=\frac{3F_{LOP}L}{2bh_{sp}{}^{2}}$$
(2)$$f_{Rj}=\frac{3F_{j}L}{2bh_{sp}{}^{2}}$$
where *f_LOP_* is the flexural stress that corresponds to the LOP (N/mm^2^), *F_LOP_* is the load that corresponds to the limit of proportionality (LOP) (N), *L* is the span length (mm), *b* is the width of the specimen (mm), *h_sp_* is the distance between the the lowest point of the notch and the top of the specimen (mm), *f_R.j_* is the residual flexural tensile strength that corresponds to the specified CMOD, and *F_j_* is the flexural load that corresponds to the specified CMOD. 

Figure 5 shows typical load-CMOD curves according to EN-14651 standard. *F_LOP_* can be defined in two ways, as represented by Case 1 and Case 2 in the figure. The flexural load *F_LOP_* can be determined at the CMOD at 0.05 mm (Case 1 in the figure) or by taking the highest load values in a range within 0.05 mm (Case 2). 

Table 2 summarizes the test results of the CMOD curves presented in Figure 4. 

Strain-softening characteristics were observed for the 64-NTS-0.5 after initial cracking. However, Strain-hardening was shown to occur continuously for the 64-NTS-1.0 specimen with an increase in CMOD after initial cracking. Strain-hardening characteristics were also observed for both 64-HTS-0.5 and 64-HTS-1.0. Furthermore, no abrupt load drop occurred after the specimens reached their maximum flexural strengths.

Depending on the fiber content, the flexural behavior of the specimens with an aspect ratio of 80 and a tensile strength of 1100 MPa was similar to that of specimens with an aspect ratio of 64 and a tensile strength of 1200 MPa. At the fiber content of 0.5%, the load decreased as the CMOD increased after initial cracking. At a fiber content of 1.0%, the load increased continuously after the initial crack and then decreased. The specimens with aspect ratio of 80 and tensile strength of 1600 MPa showed the best flexural behavior in this study. The specimens with 0.5% and 1.0% fiber contents maintained over 3 mm CMOD with no decrease in flexural loading.

The maximum flexural stress of the specimens with 0.5% steel fiber and a tensile strength of 1600 MPa was 20.2%–63.1% lower than that of the other specimens with a mix ratio of 1.0 percent fiber. However, a high level of residual bending stress remained after initial cracking. Therefore, the use of steel fiber with a high tensile strength, and a high aspect ratio is expected to provide excellent flexural behavior for high-strength concrete, even if the concrete has a low fiber content.

Figure 6 shows the failure surface of the CMOD specimens. Two failure modes were observed with the pull-out and/or rupture of fibers. For the specimen 80-HTS-1.0, when increasing the fiber content and tensile strength of the fiber, the majority of the fiber was pulled out from the matrix. 

### 3.2. Evaluation of Residual Flexural Strength of SFRC 

The flexural performance of the SFRC specimens was evaluated in accordance with EN-14561 in the International Federation for Structural Concrete (fib) model code 2010 (fib MC 2010) [13] for this study. The steel fiber can be substituted for tensile reinforcement for all and/or partial reinforcement when Equations (3) and (4) are satisfied:(3)fR1fLOP≥0.4
(4)fR3fR1≥0.5
where *f_R1_* is the residual flexural tensile strength in the serviceability limit state that corresponds to the CMOD at 0.5 mm (N/mm^2^), and *f_R3_* is the ultimate residual flexural tensile strength that corresponds to the CMOD at 2.5 mm.

The 80-NTS-1.0 specimen has the highest *f_R1_* value with 14.36 MPa, and the 80-HTS-1.0 specimen has the highest *f_R3_* value with 16.55 MPa. The fiber aspect ratio affects the residual flexural tensile stress up to the service load stage, and then the tensile strength of the steel fiber becomes the dominant parameter to determine the residual flexural tensile stress at the ultimate loading stage. 

Even if the number of crosslinking fibers is increased as the crack propagates, the fibers are pulled out or broken when the fiber tensile strength is low. However, the crack will continue to develop without failure when high tensile strength fiber is used. 

Figure 7 shows the effects of fiber content on the residual flexural performance of the SFRC specimens for the evaluation that includes the experimental results obtained from this study compared to those found in the literature. Concrete with strength values higher than 60 MPa was considered a high-strength concrete. For the conventional normal-strength concrete, some specimens with low fiber content (0.25 percent) did not satisfy Equation (3), and for the high-strength concrete, some specimens with 1.0% fiber did not satisfy Equation (4). The fiber tensile strength is lower than the adhesion force between the materials, even if the fiber content is high, so the flexural performance is limited due to the rupture of the fibers.

Figure 8 shows the effects of the fiber aspect ratio on the flexural performance of the SFRC specimens. In the literature, the fiber aspect ratios range from 50 to 95. The specimens having fiber with an aspect ratio of 50 did not satisfy the criteria according to Equation (3), and the specimens with fiber aspect ratios of 60 to 80 did not satisfy Equation (4).

Figure 9 presents the effects of the tensile strength of the steel fiber on the residual flexural performance of the SFRC specimens. In the literature, tensile strength values of steel fiber range from 1000 MPa to 1200 MPa, and only a limited number of studies have used high-strength steel fiber with over 1600 MPa. In this study, when the tensile strength of the steel fiber was below 1200 MPa, many specimens were disqualified based on Equations (3) and (4). However, all the specimens that contained high tensile strength steel fiber satisfied both Equations (3) and (4). For the specimens with steel fiber tensile strength values below 1200 MPa, the ratio *f_R3_*/*f_R1_* tended to decrease compared to *f_R1_*/*f_LOP_*.

Figure 10 presents the flexural performance of the SFRC specimens according to the concrete’s compressive strength and fiber content. In the case of the compressive strength of concrete at 40 MPa, most of the specimens are shown to satisfy Equation (3), except for the specimens containing 0.25% steel fiber. The *f_R1_*/*f_LOP_* of the SFRC in Figure 10a tends to increase with an increase in compressive strength. However, as the compressive strength increases, Equation (4) cannot be satisfied by some of the specimens, and the *f_R3_*/*f_R1_* value of the SFRC in Figure 10b tends to decrease.

The experimental results indicate that all the test specimens satisfy the *f_R1_*/*f_LOP_* value regardless of the tensile strength of the steel fiber. However, the *f_R3/_f_R1_* values range from 0.34 to 1.56, which do not satisfy the criteria, except for the specimens with a fiber tensile strength of 1600 MPa. Therefore, the results indicate that steel fiber with a tensile strength of 1600 MPa can be used as a feasible substitute for tensile reinforcement in high-strength concrete.

In the literature, a reinforcing index (RI) has been used to quantify the flexural performance of SFRC. The RI is computed by multiplying the fiber content (*V_f_*) and aspect ratio (*l/d*). In this study, the tensile strength of steel fiber also was determined to be an important parameter for the flexural performance of SFRC. Therefore, a modified RI that includes the tensile strength of steel fiber is proposed in this study. The modified RI value is determined by multiplying *V_f_*∙(*l/d*)∙*f_ft_*/100.

Figure 11 shows the relationship between the flexural performance of the SFRC specimens and modified RI values. The SFRC performance index *f_R1_*/*f_LOP_* in Figure 11a tends to increase with the newly proposed RI value, but the increment in the computed value (*f_R3/_f_R1_*) in Figure 11b is reduced and tends to be irregular.

### 3.3. Determination of Fracture Energy of SFRC

The fracture energy of high-strength concrete can be computed using Equations (5) and (6), as proposed by the Japan Concrete Institute (JCI-S-001-2003) [14]:(5)$$G\left(f\right)=\frac{0.75\;W_{0}+W_{1}}{A_{lig}}$$
(6)$$W_{1}=0.75\left(\frac{S}{L}m_{1}\right)g\cdot CMOD_{c}$$
where *G(f)* is the fracture energy (kN/m), *W_1_* is the work done by the applied load and self-weight of the specimen (kN·m), *W_0_* is the energy represented by the area of the load-CMOD curves (kN·m), *S* is the span length (mm), *L* is the overall length of the specimen (mm), *m_1_* is the mass of the specimen (kg), *g* is the acceleration due to gravity (m/s^2^), *CMODc* is the CMOD value at failure (mm), and *A_lig_* is the cross-sectional area of the specimen (mm^2^).

Fracture energy refers to the energy at which the material causes a fracture. Köksall et al. (2012) evaluated the fracture energy of high-strength concrete according to steel fiber content and tensile strength. Figure 12 shows the relationship between the fracture energy of the SFRC specimens used in this study and the modified RI in terms of steel fiber characteristics and fiber content. The results indicate that the fracture energy increases with an increase in fiber content and tensile strength.

As shown, the fracture energy increased by 32.5%~138.5%, with an increase in fiber content. As the fiber tensile strength and aspect ratio increased, the fracture energy increased by 194.0%~443.5% and 45.2%~140.0%, respectively. The fracture energy was influenced predominantly by the fiber’s tensile strength. In short, as described for the residual bending stress and flexural behavior evaluations, when the tensile strength of the steel fiber is sufficient, the steel fiber does not break, even if cracking occurs. The steel fiber provides high fracture energy until failure due to its continuous bridging effect.

## 4. Conclusions

This study evaluated the effects of the aspect ratio and mechanical properties of steel fiber on the flexural performance of high-strength concrete in accordance with the EN-14651 standard and JCI code. The conclusions are based on limited experimental results and are summarized as follows.

The flexural behavior in LOP was found to be influenced by the fiber content and aspect ratio. The maximum value was 8.81 MPa for the 80-NTS-1.0 specimen.

In this study, the maximum flexural stress of the SFRC specimens was found to be affected by the fiber properties in the order of fiber content, tensile strength, and aspect ratio. The maximum flexural stress was the highest at 16.86 MPa in specimens with an aspect ratio of 80 and a tensile strength of 1600 MPa with 1.0% steel fiber.

The tensile strength of steel fiber is a dominant parameter that can be used to determine the flexural performance of SFRC. The proposed modified RI indicates that high tensile strength steel fiber can be used in concrete with high compressive strength.

## Figures and Tables

**Figure 1 materials-12-02105-f001:**
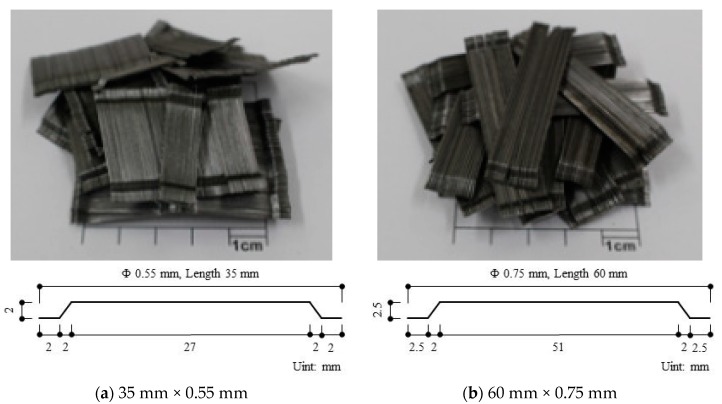
Fibers used in this study.

**Figure 2 materials-12-02105-f002:**
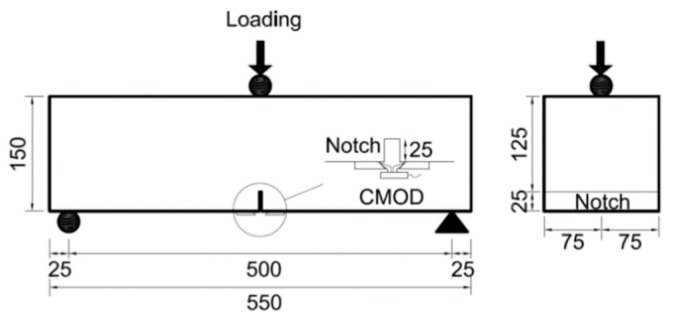
Dimensions of the beam specimen (unit: mm).

**Figure 3 materials-12-02105-f003:**
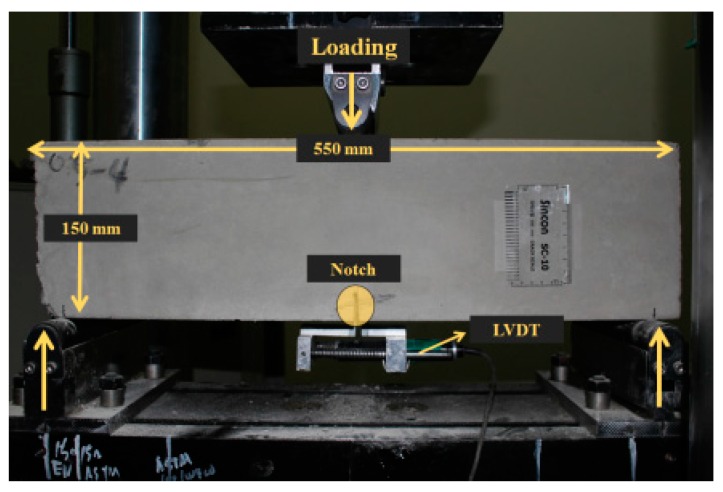
Test set-up for crack mouth opening displacement (CMOD) tests of steel fiber-reinforced concrete (SFRC) specimens.

**Figure 4 materials-12-02105-f004:**
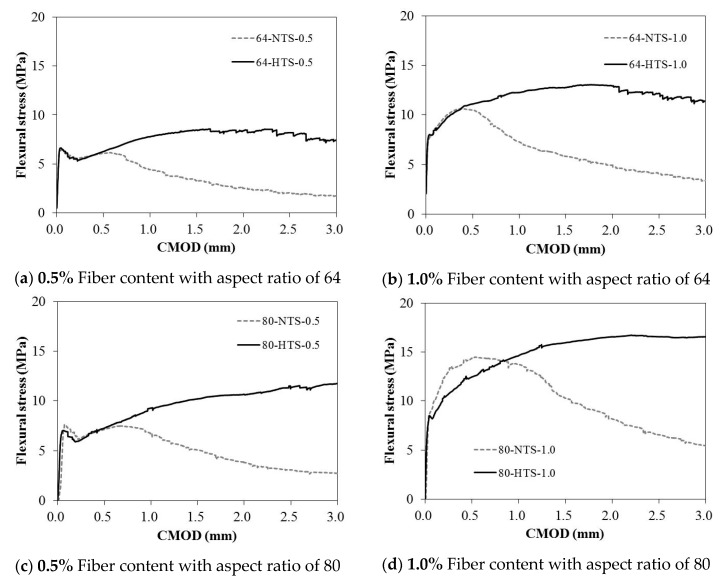
Flexural stress versus CMOD curves for high-strength SFRC specimens with different fiber contents and aspect ratios.

**Figure 5 materials-12-02105-f005:**
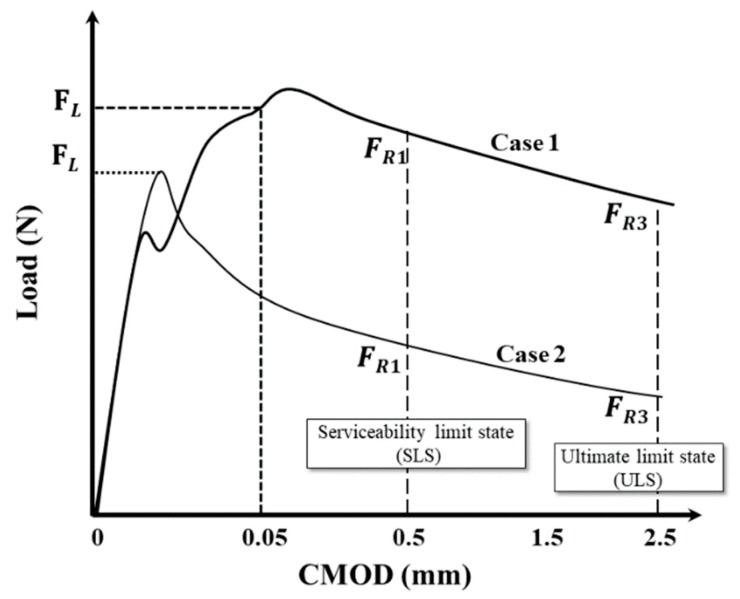
Typical load-CMOD curves (EN-14651).

**Figure 6 materials-12-02105-f006:**
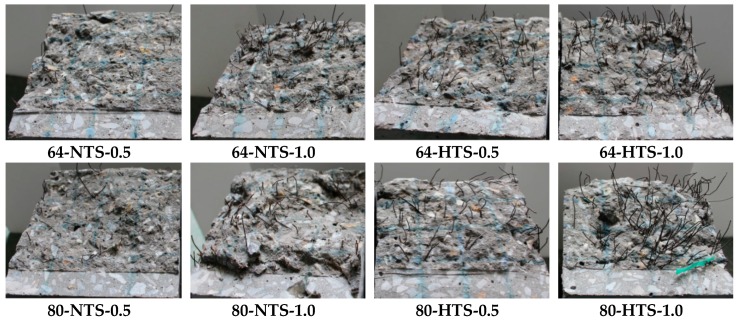
Failure modes of steel fibers in the specimens.

**Figure 7 materials-12-02105-f007:**
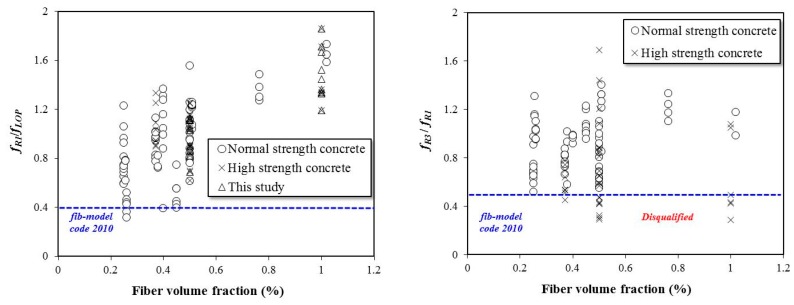
Effects of steel fiber content on the residual flexural performance of SFRC specimens.

**Figure 8 materials-12-02105-f008:**
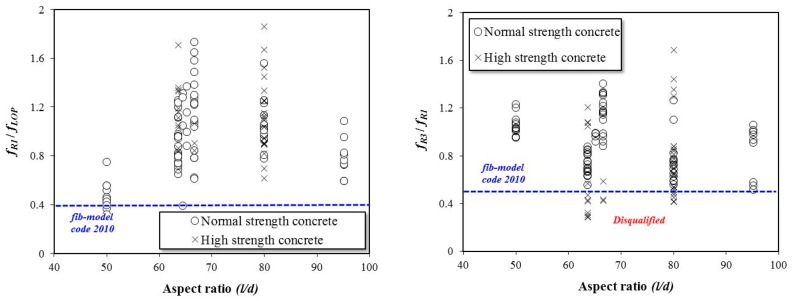
Effects of steel fiber aspect ratio on the residual flexural performance of SFRC specimens.

**Figure 9 materials-12-02105-f009:**
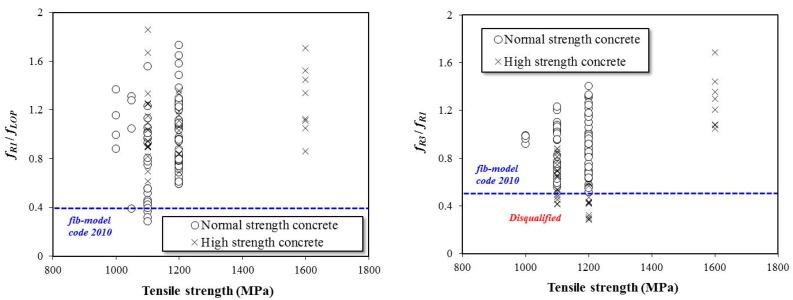
Effects of steel fiber tensile strength on the flexural performance of high-strength SFRC specimens.

**Figure 10 materials-12-02105-f010:**
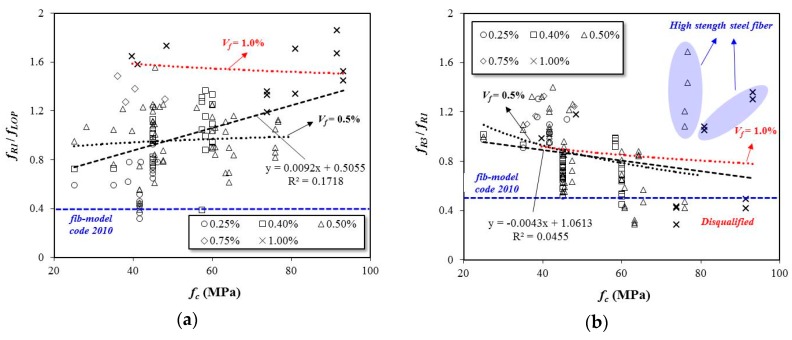
Effects of compressive strength on flexural performance of SFRC specimens, (**a**): Performance index of *f_R1_*/*f_LOP_* versus compressive strength of concrete (**b**): Performance index of *f_R3_*/*f_R1_* versus compressive strength of concrete.

**Figure 11 materials-12-02105-f011:**
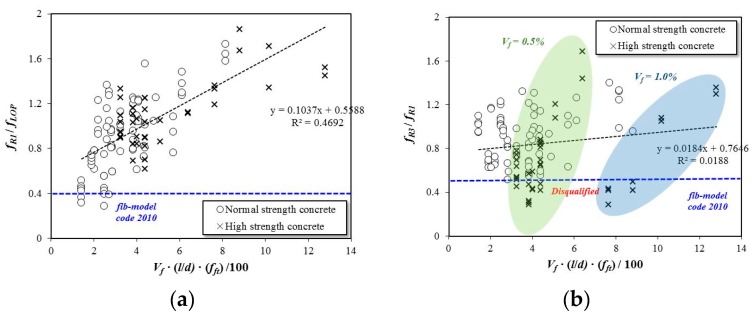
Reinforcing index versus flexural performance values. (**a**): Performance index of *f_R1_*/*f_LOP_* versus modified RI. (**b**): Performance index of *f_R_**_3_*/*f**_R1_* versus modified RI.

**Figure 12 materials-12-02105-f012:**
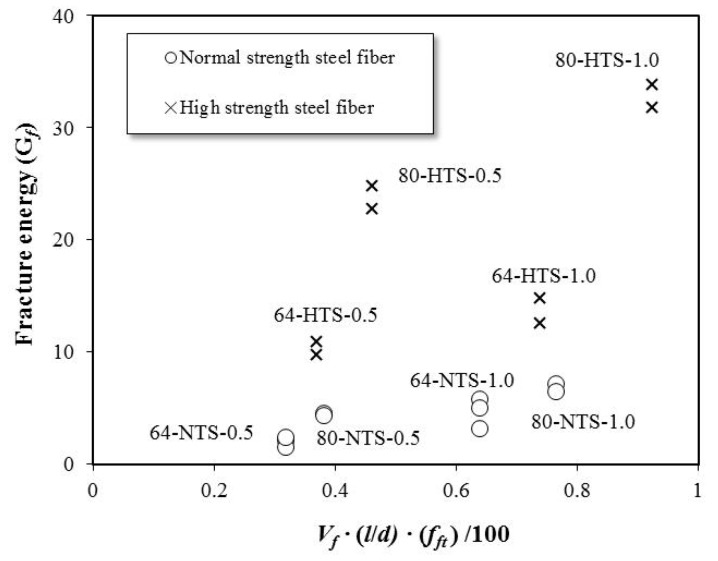
Relationship between fracture energy and the modified reinforcing index.

**Table 1 materials-12-02105-t001:** Mix design of study steel fiber-reinforced concrete.

f_ck_ (MPa)	W/B (%)	Air (%)	Unit Weight (kg/m^3^)						
			W	C	SF	FA	S	G	Steel Fiber
80	25	4	165	462	66	132	643	813	0
									39.3
									78.5

Notes: *f_ck_*: compressive concrete strength, W/B: water-to-binder ratio, W: water, C: Portland cement Type I (Density: 3.15 g/cm^3^), SF: silica fume, FA: fly ash (2.12g/cm^3^, Blaine: 2,976 cm^2^/g), S: sand (Density: 2.56 g/cm^3^, Absorption Ratio: 1.18%), G: gravel (Density: 2.65 g/cm^3^).

**Table 2 materials-12-02105-t002:** Test Results for Flexural Performance of High-Strength SFRC Mixtures.

Specimen	*f_LOP_*	*f_R_*	*f_R1_*	*f_R3_*	*f_R1_*/*f_LOP_*	*f_R3_*/*f_R1_*	*G_f_*
	(MPa)	(MPa)	(MPa)	(MPa)			(N/m)
CON	7.02	7.02	-	-	-	-	-
	(±0.29)	(±0.29)					
64-NTS-0.5	6.43	6.87	6.07	2.05	0.94	0.34	1950.70
	(±1.30)	(±0.57)	(±3.64)	(±1.13)	(±0.18)	(±0.01)	(±370.87)
64-NTS-1.0	7.83	10.69	10.50	4.09	1.34	0.39	4651.73
	(±0.50)	(±0.28)	(±0.58)	(±2.18)	(±0.07)	(±0.07)	(±1088.41)
64-HTS-0.5	6.57	8.89	6.26	8.18	0.96	1.31	10,317.59
	(±0.30)	(±0.62)	(±1.56)	(±1.79)	(±0.10)	(±0.06)	(±618.37)
64-HTS-1.0	7.98	13.06	11.09	12.18	1.39	1.10	13,673.07
	(±2.18)	(±1.53)	(±3.82)	(±2.40)	(±0.19)	(±0.01)	(±1127.82)
80-NTS-0.5	7.86	8.20	7.13	3.12	0.88	0.44	4372.64
	(±0.16)	(±0.08)	(±1.22)	(±0.06)	(±0.04)	(±0.03)	(±74.61)
80-NTS-1.0	8.81	14.50	14.36	6.55	1.63	0.45	6755.17
	(±0.77)	(±0.49)	(±1.88)	(±2.63)	(±0.10)	(±0.04)	(±344.22)
80-HTS-0.5	7.00	11.85	7.27	11.30	1.04	1.56	23,765.02
	(±0.41)	(±0.41)	(±0.15)	(±0.61)	(±0.08)	(±0.12)	(±1041.55)
80-HTS-1.0	8.39	16.86	12.46	16.55	1.49	1.33	32,813.33
	(±0.30)	(±0.39)	(±0.13)	(±0.53)	(±0.04)	(±0.03)	(±994.72)

Note: CON: control specimen; *f_LOP_* is the flexural stress that corresponds to the LOP (N/mm^2^).

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
