# Peer review of "The Influence of Steel Fiber Tensile Strengths and Aspect Ratios on the Fracture Properties of High-Strength Concrete"

_materials, 2019, doi:10.3390/ma12132105_

Round 1

Reviewer 1 Report

Very good manuscript. 

Here are few points, which i found:

Page 2, Table 1: More information about the raw materials, what kind of cement, fly ash, silica fume, sand and gravel with technical standard

Page 3: Please give more fundamental information about the geometry of the steel fibre, maybe a picture with dimensions! It is nessary because the fibres will be activated at first and after it deformed a specially in hooked area (see µXCT measurements https://doi.org/10.1016/j.conbuildmat.2019.03.227) and than cracked. The mode of action of the fibres has to explain in an better way, maybe with other references.

Page 4: How many beams you use for every mix design

Page 5: Can you give the formula for Gf? I found the formula in page 9. Maybe you give here a link.

Page 7: Where do you have the single points in the graphs? The High Performance Concrete with triangles are own values? Maybe you can show it easyer for the reader with the text below the pictures

Page 10: You write, that the tensile strength is the dominat parameter. Can you show maybe in pictures, that the fibres were cracked and not only deformed? What other influence exits. Maybe you can explain it with other references. Maybe for example the curvature has also a very high influence.

Author Response

Please refer the attached file.

Reviewer 2 Report

The work deals with a subject of current relevance in the field of fiber-renforced cememntitious composites. Specifically, it investigates the role of aspect ratio and steel strength in FRCC whose matrix is made of High-Strength Concrete.

The text is generally clear and the manuscript structure conforms to common standards in scientific writing. Therefore, the current version can be accepted after minor revision.

The Authors are invited to address the comments reported throughout the marked manuscript attached at the present e-mail. Moreover, they are requested to check the graphical quality of figures and graphs and, if needed, replace the current ones with new and sharper figures.

Author Response

Please refer the attached file.

Reviewer 3 Report

The paper titled ‘ Influence of Steel Fiber Tensile Strengths and Aspect Ratios on Fracture Properties of High-Strength Concrete’ discusses the flexural performance of SFRC specimens made with steel fibres of variable aspect ratio, tensile strength etc. The performance is evaluated using displacement tests and the results are compared with relevant codal provisions.

The paper is well written and is understandable. However, the paper does not have critical literature survey, which is why the novelty of work is not established.

1.     Line 32: Change to ‘must also be considered’.

2.     Line 78-79: Why is the tensile strength of fibres not kept constant at two aspect ratios? It would have been easier to compare the values at same strength levels.

3.     What is the least count of LVDTs? How many LVDTs are installed on each specimen to gat CMOD? Also, the test setp is not clear with respect to taking readings.

4.     Line 140-142: What is the technical reason for this behaviour?

5.     Section 3.3: More discussion on the importance of fracture energy is required.

Author Response

Please refer the attached file.
